# OpenReview forum: "Rule2DRC: Benchmarking LLM Agents for DRC Script Synthesis with Execution-Guided Test Generation"
_ICML.cc/2026/Conference — ICML 2026 regular_

### Official Review · Reviewer_jyRn · 2026-03-03

**Soundness:** 3
**Presentation:** 1
**Significance:** 3
**Originality:** 2
**Overall Recommendation:** 4
**Confidence:** 2

**Summary:**

This paper introduces Rule2DRC, a large-scale, execution-based benchmark for converting natural language design rules into KLayout DRC scripts, comprising 1000 rule-to-script conversion tasks and 13921 evaluated layouts. Furthermore, this paper proposes SplitTester, a test agent that clusters candidate scripts based on execution behavior and iteratively generates discriminative layout tests to differentiate previously indistinguishable candidate scripts, thereby improving best-of-N selection. Experiments on multiple LLMs demonstrate that SplitTester has an advantage over improved existing best-of-N test agents. But due to some drawbreaks, I can not recommend it for acceptance.

**Compliance With Llm Reviewing Policy:**

Affirmed.

**Final Justification:**

The authors have addressed my question, although there are some typos in the manuscripts, which will not impact the quality.

**Key Questions For Authors:**

- On average, how many evaluation layouts are included per task? Calculated in percentiles (e.g., median, interquartile range)? How do you assess that this number is sufficient to distinguish scripts that are semantically different but behave similarly?

- Can you provide an execution time/cost profile for each task of each test agent (e.g., average DRC executions, GDS generation counts, token usage, actual runtime), and how these metrics vary with N and the test budget?

- How frequently do the test expectation labels generated by LLM become incorrect? How sensitive is the choice of SplitTester to this noise? (e.g., how much would performance change if expectation labels were removed and only clustering and judgments were relied upon?)

- Can you break down performance by rule category (SkyWater derived rules, synthetic multi-constraint rules, and syntax coverage rules) to better understand in which aspects execution bootstrapping provides the greatest benefit?

- Do you plan to release metadata for each task (e.g., syntax operators used, number/type of boundary cases) and a small hidden test set to prevent overfitting? If open source, how will you maintain the integrity of "unknown" evaluation layouts?

**Limitations:**

yes

**Strengths And Weaknesses:**

## Strengths

- A larger-scale NL to DRC translation benchmark is introduced, and an execution-based evaluation method is adopted to address the shortcomings of previous work, which mainly relied on code similarity metrics and small-scale non-open datasets.

- This includes ablation analysis of the SplitTester components (judging LLM, clustering scoring criteria, and curators), and a contextual engineering study demonstrating the significant benefits of providing KLayout files.

- Clear formulas, success/error metrics, and an algorithmic description of SplitTester are presented, along with easily understandable pseudocode.

## Weaknesses

- Correctness is defined as complete consistency with the reference output on the provided evaluation layout; while a reasonable number of layouts per task (approximately 14 on average) may not be sufficient to guarantee semantic equivalence across the entire rule space, especially for complex or special cases.

- Early scoring in SplitTester relies on test labels generated by LLM, which introduces potential label noise. Although the final judging phase mitigates this, it still affects the selection of clustering targets.

- No runtime or cost analysis (actual runtime, number of DRC runs per task, token usage) of the test agent is provided, which is crucial for agent methods that repeatedly execute code and generate GDS layouts.

- No connection to broader execution bootstrapping or selection-time code generation methods (e.g., EG-CFG, CodeTree, AutoTest) is discussed; placing SplitTester within these common techniques strengthens its position.

- Obvious spelling errors, such as “surface-level code smilarity”, “translating high-level NL design rules into correct, executable DRC script”, “with 1,000 rule translation task”, etc.

---

> ### Author Rebuttal · Authors · 2026-03-31
>
> Thank you for the valuable feedback. We address your questions and concerns below.
>
> ---
> **Q1. What is the average number of evaluation layouts included per task, and what are its percentiles? How do you assess that this number is sufficient to distinguish scripts that are semantically different but behave similarly? (W1, Q1)**
>
> We report the distribution statistics (average and quantiles) of evaluation layouts, method calls, and unique methods per task category in Table A3.
>
> To ensure a sufficient number of test cases for each task, we systematically include threshold and topological corner cases for each operation used in the design rule. For details on annotation cost and the quality assurance process, please refer to our **response to Q1 from Reviewer PLgw.**
>
> *Table A3: https://drive.google.com/file/d/1MAFGKcHJYKZxF5IN8be4D3l4266mFkgQ*
>
> ---
> **Q2. How frequently do the test expectation labels generated by LLM become incorrect? Although the final judging phase mitigates this, it still affects the selection of clustering targets. How much would performance change if expectation labels were removed and only clustering and judgments were relied upon? (W2, Q3)**
>
> Table A4 reports the label error rates of LLM-generated expected labels relative to ground-truth DRC script outputs. The label error rate ranges from 15% to 37%, with stronger models showing lower error rates. As noted, SplitTester's final judging phase helps mitigate these errors.
>
> To analyze the contribution of each component, we present the results of two ablated variants in Table A5: (1) No final judge, which relies solely on expectation labels and returns the top-1 script without the final judging, and (2) No expected labels, which relies only on cluster size to select the target cluster and top-3 elements to show the judge, without using expected labels. Both variants lead to notable performance drops compared to SplitTester, indicating that the two components are each insufficient alone but serve complementary roles. We will include these results in the revision.
>
> *Table A4: https://drive.google.com/file/d/1B6T2lvelGuZdEZCUbDrLQQltJ1LQt7Zk*
>
> *Table A5: https://drive.google.com/file/d/1QfNEnWIMgouUW0x5TTac-qdo4u7FVAl4*
>
> ---
> **Q3. No runtime or cost analysis of the test agent is provided, which is crucial for agent methods that repeatedly execute code and generate GDS layouts. (W3, Q2)**
>
> Please refer to our **response to Q3 from Reviewer PLgw,** where we present a detailed cost breakdown (runtime, number of DRC runs, token usage) for evaluating the agents on the full 1,000 tasks of the Rule2DRC benchmark and discuss the tradeoffs of our method.
>
> ---
> **Q4. No connection to broader selection-time code generation methods is discussed. (W4)**
>
> In Table A6, we extend our tester agent to an iterative code refinement setting, where the model improves BoN-selected code over up to $M=3$ rounds, accepting revisions only if they improve collected test case scores. This monotonically improves performance, with SplitTester outperforming Generated Tests, the only other baseline generating expected outputs, in success rate, while matching it in error rate. We will expand these experiments in the revision.
>
> *Table A6: https://drive.google.com/file/d/1KArPjG73wzRPeB6863lJAmaa9zvrv5I2*
>
> ---
> **Q5. Spelling errors. (W5)**
>
> Thank you for pointing these out. We will fix all typos and spelling errors in the revision.
>
> ---
> **Q6. Can you break down performance by rule category to better understand in which aspects execution bootstrapping provides the greatest benefit? (Q4)**
>
> In Table A7, we provide a performance breakdown by rule category (SkyWater-derived, synthetic multi-constraint, and syntax coverage rules) for GPT-OSS-120B across all Best-of-$N$ settings. The results show that SplitTester consistently improves upon the baselines across all categories and Best-of-$N$ settings. Notably, the benefit is greater in the more challenging synthetic multi-constraint category, where each task involves multiple constraints and operators. We will include these results and related discussion in the revision.
>
> *Table A7: https://drive.google.com/file/d/1lDwBuOkgBK5SerC7GlrsYXvHOH-mN5qj*
>
> ---
> **Q7. Do you plan to release metadata for each task (e.g., syntax operators used, number/type of boundary cases) and a small hidden test set to prevent overfitting? If open source, how will you maintain the integrity of "unknown" evaluation layouts? (Q5)**
>
> Thank you for the suggestion. We plan to release metadata (including the statistics reported in Table A3), the evaluation pipeline, and the benchmark to lower the barrier for future research in this domain. To address the risk of benchmark overfitting, we will additionally construct 100 private tasks following the identical procedure and maintain a public leaderboard tracking model performance. By comparing results on the public and private splits, we will measure the degree of overfitting and monitor it as new models are released.

---

> > ### Author Rebuttal · Reviewer_jyRn · 2026-04-03
> >
> > Thanks for your response. I will change my score from 3 to 4.

---

> > > ### Author Response · Authors · 2026-04-06
> > >
> > > Thank you for your response and for increasing your score. We truly appreciate the time and effort you have spent providing valuable feedback.

---

### Official Review · Reviewer_PLgw · 2026-03-06

**Soundness:** 2
**Presentation:** 2
**Significance:** 3
**Originality:** 3
**Overall Recommendation:** 4
**Confidence:** 4

**Summary:**

The paper introduces Rule2DRC, a benchmark for translating natural-language design rules into executable Design Rule Check (DRC) scripts, consisting of 1,000 tasks and 13,921 evaluation layouts evaluated via execution equivalence against ground-truth scripts. The authors also propose SplitTester, a tester module that iteratively generates discriminative layout tests, clusters candidate scripts by execution behavior, and selects the best candidate under a best-of-N generation setting. Experiments indicate that execution-guided test generation improves script selection accuracy compared to existing tester strategies.

**Compliance With Llm Reviewing Policy:**

Affirmed.

**Final Justification:**

The authors provided extensive additional experiments and clarifications that successfully addressed my concerns. These updates significantly strengthen the paper's claims, justifying an upgrade of my recommendation to 4.

**Key Questions For Authors:**

Q1 - Benchmark Construction and Validation.
How were synthetic rules generated and validated to ensure realism and practical relevance? What procedures were used to verify the correctness of ground-truth scripts and associated layout outputs? Providing details on annotation cost and quality assurance would strengthen confidence in the benchmark.

Q2 - Semantic Correctness and Evaluation Robustness.
How are cases where a generated script compiles and executes successfully, but fails to detect all intended violations or enforces an incorrect interpretation of the design rule handled? In particular, does the evaluation protocol explicitly account for such semantic errors beyond compile-time and runtime failures?

Q3 - Computational Efficiency.
What is the computational overhead of SplitTester relative to baseline selection strategies, measured in terms of wall-clock time or overall compute cost? For example, how many and which types of GPUs were used in the experiments? Were any efficiency strategies employed during LLM execution?

Q4 - Sensitivity to Test Budget.
SplitTester relies on a test budget parameter. How sensitive is performance to this parameter, and does effectiveness degrade significantly under smaller budgets?

The rebuttal will positively affect my evaluation if it clarifies the benchmark construction process, provides stronger justification of the evaluation protocol (particularly regarding the semantic correctness of generated scripts), and reports the computational overhead and sensitivity analyses of SplitTester.

**Limitations:**

Yes.

**Strengths And Weaknesses:**

Strengths
S1 - Domain-Specific Benchmark and Open Resources.
The introduction of a DRC-specific benchmark addresses a largely underexplored and traditionally proprietary domain. The release of the dataset and evaluation code lowers barriers to research in AI-assisted electronic design automation (EDA). The use of execution-based evaluation rather than surface-level code comparison strengthens the benchmark’s methodological rigor.

S2 - Application to a Specialized and Practically Relevant Task.
The formulation of DRC script generation as an executable code-generation problem is well motivated. The proposed evaluation framework enables scalable experimentation and may extend to additional design rules or process design kits (PDKs), suggesting broader applicability.

Weaknesses
W1 - Limited Transparency in Task Construction.
The process for constructing ground-truth scripts and associated layouts lacks sufficient detail. Manual generation of nearly 14K layouts appears labor-intensive, yet no discussion of annotation cost or effort is provided. The generation and validation of synthetic rules are insufficiently described, making it difficult to assess their realism and practical relevance. Since the benchmark is a primary contribution of the paper, additional transparency and reproducibility details would further strengthen it.

W2 - Evaluation Metrics and Result Interpretation.
The evaluation primarily reports success rate and error rate, while improvements over baselines appear relatively modest. Additional metrics (such as compute cost or time-to-best-candidate) would help contextualize the practical trade-offs of the proposed selection strategy. Furthermore, it would be useful to clarify how the evaluation handles cases where scripts compile and execute successfully but partially enforce the intended rule, as this affects whether semantic correctness is fully captured.

W3 - Limited Sensitivity in Ablation Study.
The ablation results suggest limited performance sensitivity to certain components. Some variants show minimal degradation. This weakens the empirical evidence that the proposed target cluster scoring mechanism is critical to overall performance.

---

> ### Author Rebuttal · Authors · 2026-03-31
>
> Thank you for the valuable feedback. We address your questions and concerns below.
>
> ---
> **Q1. How were synthetic rules generated and validated to ensure realism and practical relevance?  What procedures were used to verify the correctness of ground-truth scripts and associated layout outputs? (W1, Q1) (Reviewers XrUj, PLgw, jyRn)**
>
> We detail the annotation costs and quality assurance process below.
>
> * **SkyWater-derived rules (2 months of manual effort).** We extracted natural language rules from the public PDK, implemented KLayout DRC scripts, and manually constructed GDS test layouts. To ensure coverage, we systematically designed hard corner test cases based on two principles for each operation:
>   - *Threshold corner cases,* where layouts pass/fail at the minimum resolution (1 nm) of the rule threshold (e.g. 4.999 µm (Fail) / 5.000 µm (Pass) / 5.001 µm (Pass)  for a 5.0 µm minimum separation)
>   - *Topological corner cases,* where test layouts cover distinct spatial relationships (e.g., partial overlap, full containment, no overlap).
>
> * **Synthetic rules (1 additional month of effort).** We used the GPT 5.2 Thinking-High API to draft initial tasks, DRC scripts, and test cases, grounding them in the verified SkyWater-derived tasks above. The model was instructed to reuse the same core layers used in real PDK tasks, while introducing multi-constraint rules or niche grammar, and to add all types of corner test cases for each operation enforced in the rule and DRC script. Authors with domain expertise then manually reviewed all DRC scripts and tasks, correcting any misalignment between scripts and rules, and ensuring that they are not only syntactically valid but also realistic and qualitatively plausible. Test cases that did not match the corrected ground-truth script output were removed, and additional corner test cases were added as needed.
>
> ---
>
> **Q2. How are cases where a generated script compiles and executes successfully, but fails to detect all intended violations or enforces an incorrect interpretation of the design rule handled? (W2, Q2)**
>
> Our evaluation explicitly accounts for such semantic errors. We report two distinct metrics: error rate (the ratio of scripts that fail to compile or execute) and success rate (correct only if outputs on test layouts exactly match the ground-truth script's outputs). A script that executes without error but misses violations or misinterprets the rule is not counted as successful in the success rate. To maximize coverage of such semantic errors, we constructed test layouts to be as comprehensive as possible, as discussed in our **response to Q1**.
>
> ---
> **Q3. What is the computational overhead of SplitTester relative to baseline selection strategies, measured in terms of wall-clock time or overall compute cost? Were any efficiency strategies employed? (W2, Q3) (Reviewers PLgw, jyRn)**
>
> Table A1 reports the total end-to-end runtime, generated tokens, and DRC evaluation counts for each method across models over all 1,000 Rule2DRC tasks. SplitTester generally incurs 5% to 40% more cost than baseline tester agents, as it continuously generates tests targeting clusters of indistinguishable candidates rather than pruning early (CodeMonkey) or testing only already-distinguished candidates ($S^*$). In return, this targeted approach yields higher success rates. For efficiency, we implemented all methods to maximally utilize prefix caching and parallelized batched LLM calls and DRC evaluations. We will include this analysis and discussion about the tradeoff in the revision.
>
> *Table A1: https://drive.google.com/file/d/1Mev5a5UyIgM84tonNX1PV5Lvrgoy3xiQ*
>
> ---
> **Q4. The ablation results suggest limited performance sensitivity to certain components. (W3)**
>
> In Table 3 of the paper (ablation studies), three variants showed minimal impact: (3) cluster selection by size, (4) by initial test scores, and (5) replacing the curator LLM with random sampling. For the alternative scoring methods ((3) and (4)), we view this insensitivity as a strength rather than a limitation, as it demonstrates robustness to the scoring function choice and confirms that the main improvement comes from generating cluster-distinguishing test cases. For (5), we chose to keep curator LLM for better scalability to larger $N$, where actively selecting diverse candidates could outperform random sampling.
>
> ---
> **Q5. How sensitive is performance to the test budget parameter, and does effectiveness degrade significantly under smaller budgets? (Q4) (Reviewers XrUj, PLgw)**
>
> Table A2 reports results under a smaller budget (4+4 tests). SplitTester consistently outperforms all baselines across models and Best-of-N settings. Its gains over tester agent baselines are even larger than under the original 8+8 budget (Table 2 of the paper) and grow with $N$, suggesting SplitTester is especially effective in more constrained, challenging settings.
>
> *Table A2: https://drive.google.com/file/d/1Ge5XrtpQX0KcJx9i4qHyU7bXyrGFkdwP*

---

> > ### Author Rebuttal · Reviewer_PLgw · 2026-04-02
> >
> > While the rebuttal provides useful clarifications, a couple of concerns raised in the review remain insufficiently addressed.
> >
> > **Limited empirical support for component importance** The rebuttal interprets weak sensitivity as robustness, but does not provide additional evidence demonstrating that the proposed components are necessary for performance gains. This leaves the causal contribution of key design choices insufficiently substantiated.
> >
> > **Metric sufficiency and interpretation** While the rebuttal clarifies that semantic correctness is captured via execution equivalence, it does not address whether this binary metric adequately reflects partial correctness or varying severity of errors. Additionally, the discussion of performance gains remains limited without a deeper contextualization of their practical significance against the increase in computational overhead.

---

> > > ### Author Response · Authors · 2026-04-03
> > >
> > > We thank you for the helpful and detailed feedback. We would like to present additional experiments that address your remaining concerns.
> > >
> > > ---
> > >
> > > **Q1. Binary metric may not adequately reflect partial correctness or varying severity of errors.**
> > >
> > > Thank you for this suggestion. To capture varying levels of severity in errors, we further measure a per-task F1 score and report the average across all 1,000 tasks for each method in *Table A8*.
> > >
> > > The per-task F1 score is defined as the harmonic mean of precision and recall of the design violation labels (violation/pass) over the GDS test cases. Unlike the binary success rate, this is a continuous score between 0 and 1 that credits partially correct solutions. A DRC script that correctly identifies most but not all violations still receives a positive score proportional to its correctness.
> > >
> > > As shown in Table A8, SplitTester consistently outperforms all baselines under this continuous metric as well. We will include these results in the revision.
> > >
> > > *Table A8: https://drive.google.com/file/d/1a-D9rd3jwUzXpvw23ZM7PMGdA-s1cPk3*
> > >
> > > ---
> > >
> > > **Q2. Practical significance against the increase in computational overhead, and necessity of proposed components.**
> > >
> > > Thank you for pointing this out. To address both the computational overhead concern and the question of component necessity, we ran additional experiments with a more efficient configuration: we reduced the early-stop parameter from $P=4$ to $P=1$ (see Appendix A.1 of the paper) and removed the curator LLM, the component that showed limited performance impact in our ablation study. We denote this variant as SplitTester (Efficient).
> > >
> > > *Figure 1* below presents the Pareto curve for each method using the GPT-OSS-20B model on the Rule2DRC benchmark, where each point corresponds to a different Best-of-N setting, the x-axis denotes end-to-end runtime (minutes), and the y-axis denotes success rate (%). The results show that SplitTester (Efficient) is Pareto-dominant over all other methods, including the original SplitTester, achieving comparable or higher success rates at lower computational cost.
> > >
> > > We will include this efficient variant in the revision along with cost-success rate plots and more extensive experiments across different models.
> > >
> > > *Figure 1: https://drive.google.com/file/d/1Xr9-sjG8-OlTFtQLOU10WTnKMj0rpvai*

---

### Official Review · Reviewer_XrUj · 2026-03-14

**Soundness:** 3
**Presentation:** 3
**Significance:** 3
**Originality:** 4
**Overall Recommendation:** 4
**Confidence:** 3

**Summary:**

The paper introduces Rule2DRC, a large benchmark for translating natural language design rules into DRC scripts. It contains 1,000 tasks and 13,921 layouts and evaluates models using execution-based correctness instead of code similarity. It also propose SplitTester, a method that generates additional test layouts to better distinguish between candidate scripts. Experiments show that SplitTester improves the success rate of LLM-generated DRC scripts.

**Compliance With Llm Reviewing Policy:**

Affirmed.

**Final Justification:**

I choose to remain initial rating since this paper presents an interesting new task for LLM coding.

**Key Questions For Authors:**

Please see weakness section.

**Limitations:**

Yes

**Strengths And Weaknesses:**

Strengths:

* The considered DRC translation problem is important and well motivated.
* The collected large-scale Rule2DRC dataset is highly valuable to future work.
* The design of SplitTester is clean and well-motivated.

Weakness

* Limited evaluation. The main evaluation is limited to one DRC engine, and scripting language. The benchmark mainly uses KLayout and its built-in DRC DSL. This limits the broad application to other intrustrial DRC ecosystem.
* Limited performance gain for SplitTester. It seems that SplitTester has few margin performance improvement over other agentic baseline.

Minor:

* Large portion of synthetic rules. A large portion of the tasks are synthetic. This may raise a bit concern about how they cover realistic DRC rules.

---

> ### Author Rebuttal · Authors · 2026-03-31
>
> Thank you for your positive review. We address below your questions.
>
> ---
>
> **Q1. The main evaluation is limited to one DRC engine, and scripting language. The benchmark mainly uses KLayout and its built-in DRC DSL. (W1)**
>
> We target KLayout because it is the only widely available open-source domain-specific language (DSL) for design rule checking (DRC). In contrast, other major DRC scripting languages such as Calibre SVRF and Cadence PVL are proprietary, which makes it difficult to release an open benchmark or support fully reproducible evaluation in those settings.
>
> We believe that providing a fully open-sourced benchmark, including data, evaluation and data generation pipelines, and source code, for this largely unexplored, proprietary domain is a valuable contribution in itself. Furthermore, our data generation pipeline, evaluation metric, and proposed tester agents are all language-agnostic and can naturally extend to other scripting languages.
>
> ---
>
> **Q2. It seems that SplitTester has few margin performance improvement over other agentic baseline. (W2)**
>
> The performance gains from SplitTester are especially prominent in more challenging settings. Specifically, our method shows clearer advantages (1) under Best-of-N with larger $N$, and (2) under lower test budgets. For results under Best-of-N with larger $N$, refer to BoN-20 in Table 2 of the paper, where the gap over baselines widens compared to smaller $N$. For results under a lower test budget (4 initial + 4 additional tests), please refer to our **response to Q5 from Reviewer PLgw,** where we show that SplitTester achieves larger gains over tester-agent baselines when fewer tests are available.
>
> Beyond these challenging settings, we highlight that SplitTester also achieves consistent improvements across diverse model types (Qwen3-30B-A3B-Instruct, GPT-OSS-20B, and GPT-OSS-120B) and across different N values in the BoN setting. We will include these discussions and results in the revision.
>
> ---
>
> **Q3. A large portion of the tasks are synthetic. This may raise a bit of concern about how they cover realistic DRC rules. (W3)**
>
> Our synthetic tasks in Rule2DRC are grounded in verified SkyWater-derived tasks extracted from a real public PDK. They reuse the same core layers and operations found in real PDK rules and are designed to capture multi-constraint settings and niche grammar, which are key characteristics of production-grade proprietary rules for advanced nodes. All synthetic tasks are then manually reviewed by authors with domain expertise for correctness, test case coverage, and practical relevance. For details on annotation cost and the quality assurance process in benchmark construction, please refer to our **response to Q1 from Reviewer PLgw.**

---

> > ### Author Rebuttal · Reviewer_XrUj · 2026-04-02
> >
> > Thanks for the detailed response. I will remain my rating.

---

> > > ### Author Response · Authors · 2026-04-06
> > >
> > > Thank you for your response and for your positive review. We truly appreciate the time and effort you have spent providing valuable feedback.

---

### Decision · Program_Chairs · 2026-04-30

**Decision:**

Accept (regular)

**Comment:**

This paper was positively received overall. Reviewers agreed that the problem is important and timely, and that Rule2DRC is a valuable contribution as a large, execution-based benchmark for a specialized domain that has previously been difficult to study in an open and reproducible way. The benchmark itself was seen as a meaningful resource, and the proposed SplitTester module was viewed as a clean and practically motivated method.

The main concerns were about benchmark transparency, cost and runtime tradeoffs, sensitivity to test budget, and the scope of evaluation beyond the current DRC engine and scripting environment.  For the final version, the authors should integrate the additional experimental details from the rebuttal, improve the presentation and fix the noted typos, and make the benchmark documentation as complete as possible, including task metadata, quality-control details, and the planned discussion of public versus private evaluation splits.